# Irregularity Reflection Neural Network for Time Series Forecasting

## Abstract

Time series forecasting is a long-standing challenge in a variety of industries, and deep learning stands as the mainstream paradigm for handling this forecasting problem. With recent success, representations of time series components (e.g., trend and seasonality) are also considered in the learning process of the models. However, the residual remains under explored due to difficulty in formulating its inherent complexity. In this study, we propose a novel Irregularity Reflection Neural Network (IRN) that reflect the residual for the time series forecasting. First, we redefine the residual as the irregularity and express it as a sum of individual, short regular waves considering the Fourier series in a micro perspective. Second, we design a module, based on the convolutional architectures to mimic the variables of the derived irregularity representation, named Irregularity Representation Block (IRB). IRN comprises IRB on top of a forecasting model to learn the irregularity representation of time series. Extensive experiments on multiple real-world datasets demonstrate that IRN outperforms the state-of-the-art benchmarks in time series forecasting tasks.

## 1 Introduction

Owing to the ubiquitous computing systems, time series is available in a wide range of domains including traffic (Chen et al., 2001), power plant (Gensler et al., 2016), stock market indices (Song et al., 2021), and so on (Liu et al., 2015; Duan et al., 2021). Spontaneously, interests in time series forecasting have grown, and as a result, an intensive research for a more accurate prediction.

In recent literature, many deep learning models have been favored for forecasting problems (Lim & Zohren, 2021). Recurrent Neural Network (RNN) and its extensions such as Long Short-Term Memory (LSTM) (Hochreiter & Schmidhuber, 1997) and Gated Recurrent Unit (GRU) (Chung et al., 2014) are popular choices for analyzing long sequences. Nevertheless, these models tend to be restricted in handling multivariate time series. As a powerful alternative, Convolu-

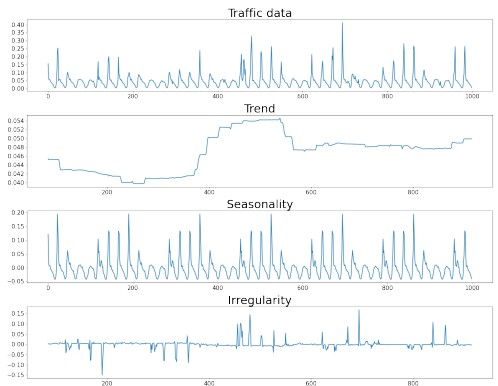

Figure 1: The Traffic data and its time series components (i.e., trend, seasonality, and irregularity).

tion Neural Networks (CNNs) has been introduced to capture overall characteristics of time series through parallel calculations and filter operations. Building on the success in forecasting task, CNN-based models have been proposed according to the type of time series data. Temporal Convolutional Network (TCN) was applied to audio datasets (Oord et al., 2016), whereas Graph Convolutional Network (GCN) was utilized in the time series with graph characteristics (e.g., human skeleton-based action recognition (Zhang et al., 2020) and traffic dataset (Bai et al., 2020)). The attention models have also been applied to emphasize the specific sequence data that are primarily referenced when making the predictions (Liu et al., 2021b).

Despite the great efforts made, forecasting performance has room for further improvement as afore-mentioned models learn feature representations directly from complex real-world time series, often overlooking essential information. Recently, incorporating representations of time series components (e.g., trend, seasonality) used in conventional econometric approaches have shown to lead to better performances of the learning models. For instance, N-BEATS (Oreshkin et al., 2019), Autoformer (Wu et al., 2021), and CoST (Woo et al., 2022) reflected the trend and seasonality of the time series and achieved improvements. However, as shown in Figure 1, time series also include the irregularity that is not accounted by the trend and seasonality, and is yet under explored (Woo et al., 2022).

To address this challenge, we show how to deal with the irregularity of the time series data to improve the forecasting performance of the deep learning models. To this end, we represent the irregularity into an encodable expression on basis of Fourier series viewed from a micro perspective. The derived representation is encoded using convolutional architectures, and named as Irregularity Representation Block (IRB). Then, IRB embedded on a base model builds the Irregularity Reflection Neural Network (IRN). We demonstrate that IRN outperforms existing state-of-the-art forecasting models on eleven popular real-world datasets.

## 2 RELATED WORK

### 2.1 DEEP LEARNING FOR TIME SERIES FORECASTING

Sequential deep learning models such as RNN, LSTM, and GRU have long been used for time series forecasting (Elman, 1990; Hochreiter & Schmidhuber, 1997; Chung et al., 2014). Although effective in capturing the temporal dependencies of time series, RNN-based models neglect the correlations in-between time series. To tackle this issue, Liu et al. (2020) propose a dual-stage two-phase (DSTP) to extract the spatial and temporal features. Shi et al. (2015) present convLSTM replacing the states of LSTM block with convolutional states. Another limitation of the sequential models are that the discrepancy between ground truth and prediction is accumulated over time as predictions are referred to predict further into the future (Liu et al., 2021a).

More recent works have demonstrated that CNNs can be applied in multivariate time series problems as well. Ravi et al. (2016) introduce the 1D convolution for human activity recognition, whereas Zhao et al. (2017) suggest the use of 2D convolution. CNN models are parallelizable, and hence show following advantages: the consideration of the correlation between variates and the prevention of error accumulation (Liu et al., 2019). A downside is the limited receptive field when predicting long sequences due to the increasing number of the parameters (Zhao et al., 2017). Wang et al. (2019) tackle this challenge by decomposing the long sequences according to long, short and closeness.

CNN-based models have received increasing attention to enhance the forecasting performance. For example, the dilated casual convolutional layer is used to increase the receptive field by down-sampling and improve long sequences prediction (Sen et al., 2019; Oord et al., 2016). Another approach is Graph Convolutional Network (GCN), that analyzes the relation between nodes with specific position and edge relation, especially in traffic data (Fang et al., 2021; Song et al., 2020) and human body skeleton data (Yoon et al., 2022; Chen et al., 2021). Attention-based models have also been adopted (Liu et al., 2019) and further developed into Transformer (Zhou et al., 2021; Liu et al., 2021b). However, these approaches do not take into account the characteristics of time series such as trend, seasonality and irregularity.

### 2.2 REFLECTING THE REPRESENTATIVE COMPONENTS OF TIME SERIES

Considerable studies on time series analysis have relied on the decomposition of time series into non-random components. For instance, DeJong et al. (1992) conducted analysis on the trends of the macroeconomic time series as well as Lee & Shen (2009) emphasized the importance of obtaining significant trend relationship in linear time complexity. Jonsson & Eklundh (2002) extracted and analyzed the seasonality of the time series data and Taylor & Letham (2018) considered both trend and seasonality. When extracting these non-random components, a non-stationary time series becomes stationary, meaning time-independent. As conventional statistical methods such as ARIMA (Autoregressive Integrated Moving Average) (Williams & Hoel, 2003) and GP (Gaussian Process)

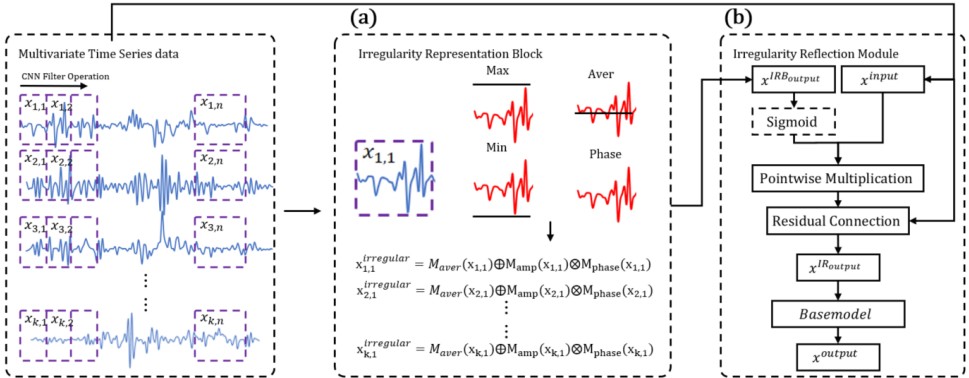

Figure 2: An overview of IRN framework. In IRN, (a) IRB extracts the irregularity feature from the input sequences and (b) Irregularity Reflection module conducts the time series forecasting.

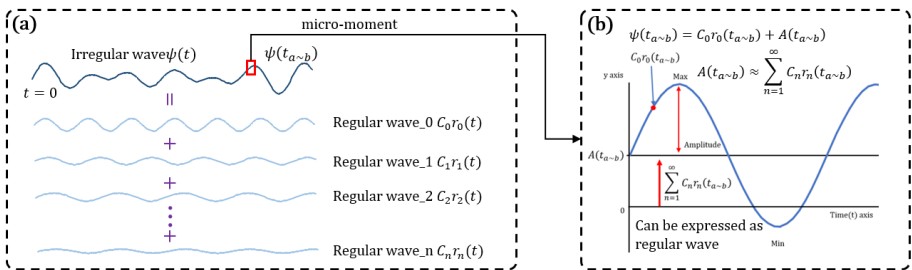

Figure 3: (a) The irregular wave $\psi(t)$ consisting of multiple regular waves and (b) the irregular wave from a micro perspective $\psi\left(t_{a \sim b}\right)$.

(Van Der Voort et al., 1996) perform better on stationary data (Cheng, 2018), differentiation for stationarity has been conducted (Atique et al., 2019). As such, extraction of the representative time series components for forecasting problems has been a major research topic (Brockwell & Davis, 2009; Cleveland et al., 1990).

Recently, direct learning from input sequences of the deep forecasting models is regarded to be enough, thereupon researchers focus on how to incorporate the components of time series in the learning process. For instance, Oreshkin et al. (2019) proposed a hierarchical doubly residual topology as the interpretable architecture to extract time series representations: trend, and seasonality. Wu et al. (2021) proposed a transformer-based model which decomposes and reflects the trend and seasonality by using auto-correlation mechanism. Woo et al. (2022) introduced disentangled Seasonal-Trend Representation Learning by using the independent mechanisms. They deviced disentanglers for the trend and seasonal features, mainly composed of a discrete Fourier transform to map the intermediate features to frequency domain. These studies successfully reflect representations of trend and seasonality which are the time dependent value, and improve forecasting performances. However, the irregularity, which cannot be explained by the trend or seasonality and is the time independent value, is not sufficiently addressed. In this paper, we build and reflect the irregularity representation to complement the previous researches in forecasting tasks.

## 3    METHODOLOGY

In this section, we discuss how to reinterpret the irregularity of the time series in term of Fourier series, extract and reflect the irregularity representation using convolutional architectures. Our proposed model IRN is shown in Figure 2.

### 3.1    THEORETICAL APPROACH

A time series is generally in the form of an irregularity. Hence, its representation is essential for time series forecasting. Among many existing approaches to represent irregularity, Fourier series is perhaps the most widely used. Fourier series approximates irregularity by the linear superposition of multiple regular waves with varying height, period, and direction as depicted in Figure 3 (a) (Bloomfield, 2004). The irregularity $\psi(t)$ can be expressed as:

$$\psi(t) = \sum_{n=0}^{\infty} C_n r_n(t) \qquad (1)$$

where $r_n(t)$ is n-th regular wave, $C_n$ is the coefficient of $r_n(t)$, $t$ is the time. The concept of infinity in Equation 1 is challenging for the learning model to handle. Therefore, we reinterpret $\psi(t)$ into an encodable equation by viewing it at the micro level. When the irregularity $\psi(t)$ in the time domain is observed at

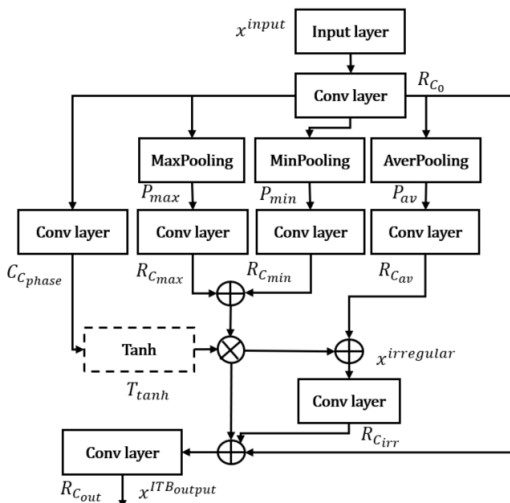

Figure 4: Architecture of the Irregularity Representation Block.

the moment $t_{a\sim b}$, it can be interpreted as a regular wave $\psi(t_{a\sim b})$ with a vertical shift, which is the average value of the regular waves. Under this concept, Equation 1 is rewritten as:

$$\psi(t_{a\sim b}) = C_0 r_0(t_{a\sim b}) + \sum_{n=1}^{\infty} C_n r_n(t_{a\sim b}) \qquad (2)$$

where $C_0 r_0(t_{a\sim b})$ is the representative regular wave characteristic. $C_0 r_0(t_{a\sim b})$ is the regular wave with a mean of 0 without vertical shift. The representative regular wave $r_0(t_{a\sim b})$ oscillates between the constant maximum and minimum values in a period of time and can be defined as $Amplitude \times sin(\omega \times t_{a\sim b})$, where the angular velocity $\omega$ is constant due to the periodicity of the wave, $\omega \times t_{a\sim b}$ is denoted as the angle $\theta$ of $r_0(t_{a\sim b})$, and $sin(\omega \times t_{a\sim b})$ is the phase of $r_0(t_{a\sim b})$. $Amplitude$ is calculated with the peaks of the wave. Accordingly, the representative regular wave $r_0(t_{a\sim b})$ can be rewritten as:

$$r_0(t_{a\sim b}) = \frac{max(t_{a\sim b}) - min(t_{a\sim b})}{2} \times sin(\theta(t_{a\sim b})) \qquad (3)$$

where $sin(\theta(t_{a\sim b}))$ is the phase of $r_0(t_{a\sim b})$ at $t_{a\sim b}$. Therefore, $C_0 r_0(t_{a\sim b})$ in Equation 2 is redefined by referring to Equation 3. The remaining infinity term $\sum_{n=1}^{\infty} C_n r_n(t_{a\sim b})$ in Equation 2 corresponds to the vertical shift and can be expressed as the average value $A(t_{a\sim b})$ of $\psi(t_{a\sim b})$ as depicted in Figure 3 (b). The representative regular wave $C_0 r_0(t_{a\sim b})$ and the average value $\sum_{n=1}^{\infty} C_n r_n(t_{a\sim b})$ convert Equation 2 into:

$$\psi(t_{a\sim b}) \approx A(t_{a\sim b}) + \frac{max(t_{a\sim b}) - min(t_{a\sim b})}{2} \times sin(\theta(t_{a\sim b})) \qquad (4)$$

When the regular waves are sequentially connected, we obtain the irregularity $\psi(t)$ consisting of the regular waves that change with time $t_{a\sim b}$. We redefine the Equation 4 as follows:

$$\psi(t) \approx A(t) + \frac{max(t) - min(t)}{2} \times sin(\theta(t)) \qquad (5)$$

where $\frac{max(t) - min(t)}{2}$ is the amplitude of the regular wave $C_0 r_0(t_{a\sim b})$ at $t_{a\sim b}$, $sin(\theta(t))$ is the phase of $C_0 r_0(t_{a\sim b})$ at $t_{a\sim b}$, and $A(t)$ is the average which is the sum of remained regular waves $\sum_{n=1}^{\infty} C_n r_n(t_{a\sim b})$ at $t_{a\sim b}$. According to Equation 5, the irregularity $\psi(t)$ can be represented by the combinations of the minimum, maximum, average, and phase values.

## 3.2 IRREGULARITY REPRESENTATION BLOCK

Based on Equation 5, the irregularity is encoded to incorporate into deep learning models. In this paper, convolutional architectures are adopted since convolutional layers allow the parallel prediction

as well as analysis of relations existing in multivariate time series through filter operations. We input the multivariate time series $x^{input} \in \mathbb{R}^{T \times d}$ , where $T$ is a look-back window of fixed length, and $d$ is the number of variates. Our model stacks multiple convolution layers with the RELU activation function, a dilation filter and same padding. The RELU activation increases the model complexity through space folding (Montufar et al., 2014), and the dilation operation helps expand the receptive fields (Oord et al., 2016). The convolution layer extracts the feature that has the same size of the $x^{input}$ through the same padding. Accordingly, Equation 5 is transformed into

$$x^{irregular} = A(x^{input}) \oplus \frac{max(x^{input}) - min(x^{input})}{2} \otimes sin(\theta(x^{input})) \qquad (6)$$

where $\oplus$ is the pointwise summation, $\otimes$ is the pointwise multiplication, and $x^{irregular}$ is the irregularity. Through this transformation, the $x^{irregular}$ is converted from time dependent to data dependent and the main operations (i.e., $A()$, $max()$, $min()$, and $sin(\theta())$) are expressed by using the convolution layers and pooling layers which extract the average, the maximum, the minimum, and the phase value from $x^{input}$ under microscopic perspective condition. The main operations are encoded like Figure 4 as follows:

$$M_{aver}(x^{input}) = R_{Cav}(P_{av}(R_{C_0}(x^{input}))) \approx A(x^{input}) \qquad (7)$$

$$M_{amp}(x^{input}) = \frac{R_{Cmax}(P_{max}(R_{C_0}(x^{input}))) - R_{Cmin}(P_{min}(R_{C_0}(x^{input})))}{2}$$
$$\approx \frac{max(x^{input}) - min(x^{input})}{2} \qquad (8)$$

$$M_{phase}(x^{input}) = T_{tanh}(C_{Cphase}(R_{C_0}(x^{input}))) \approx sin(\theta(x^{input})) \qquad (9)$$

where $P_{max}$, $P_{min}$, and $P_{av}$ are the max, min, and average pooling operations, respectively. $C$ is the convolution layer without activation, $R$ is $C$ with the RELU activation and $T_{tanh}$ is the hyperbolic tangent(tanh) activation. Through this process, the average, amplitude, and phase values in Equation 6 are converted to trainable values. To extract the representation of the average value from $x^{input}$, we stack the 2D convolution filter and the 2D average pooling as in Equation 7. To decompose the representation of the amplitude from $x^{input}$, we construct the structure same as Equation 8 with the 2D max and min pooling. To obtain the representation of the adaptive phase value using $x^{input}$ under microscopic aspect condition, referring to the Equation 9, we use the tanh activation after convolution layer. Consequently, these operations(i.e., $M_{aver}(x^{input})$, $M_{amp}(x^{input})$, and $M_{phase}(x^{input})$) extract the average, amplitude, and phase values from $x^{input}$, and we redefine Equation 6 as follows:

$$x^{irregular} = M_{aver}(x^{input}) \oplus M_{amp}(x^{input}) \otimes M_{phase}(x^{input}) \qquad (10)$$

To consider the $x^{irregular}$ value, we apply the residual stacking principle which enables complex interpretation by combining features in a hierarchical form for each step (Oreshkin et al., 2019). Therefore, we design the IRB architecture as follows:

$$x^{IRB_{output}} =$$
$$R_{C_{out}}(R_{C_{irr}}(x^{irregular}) \oplus R_{C_0}(x^{input}) \oplus M_{amp}(x^{input}) \otimes M_{phase}(x^{input})) \qquad (11)$$

The output of IRB $x^{IRB_{output}}$ is the representation of the irregularity which considers the average, amplitude, phase, and input components. Furthermore, these components are trainable values because they consist of the convolution layers.

## 3.3 IRREGULARITY REFLECTION NEURAL NETWORK

IRN consists of IRB and a irregularity reflection module as in Figure 2. For the forecasting of this study, a recent model that reflects trend and seasonality, known as SCInet (Liu et al., 2021a) is used as the base model. The $x^{IRB_{output}}$ is passed to the base model through the irregularity reflection module.

$$x^{IR_{output}} = S_{sig}(x^{IRB_{output}}) \otimes x^{input} \oplus x^{input} \qquad (12)$$

where $x^{IR_{output}}$ is the output of IRN and $S_{sig}$ is the sigmoid activation. The pointwise multiplication is applied to emphasize the irregularity of the $x^{input}$ by using the $x^{IRB_{output}}$ with $S_{sig}$. If we use the $x^{IRB_{output}}$ as the input value of the time series model, some information (e.g., trend, seasonality) can be omitted. To alleviate this problem, we preserve the original information by residual connection, which also prevents the gradient vanishing (He et al., 2016).

Table 1: Summary of datasets and evaluation metrics used for time series forecasting.

| Type | Dataset | Variates | Timesteps | Granularity | Start time | Metrics | Train/Val/Test |
|---|---|---|---|---|---|---|---|
| ETT (Zhou et al., 2021) | ETTh1 ETTh2 ETTm1 | 7 | 17420 69680 | 1 hour 15 min | 7/1/2016 | MSE MAE | 12/4/4 |
| PEMS (Chen et al., 2001) | PEMS03 PEMS04 PEMS07 PEMS08 | 358 307 883 170 | 26209 16992 28224 17856 | 5 min | 5/1/2012 7/1/2017 5/1/2017 3/1/2012 | MAE MAPE RMSE | 6/2/2 |
| Solar, Traffic, Electricity, Exchange-rate (Lai et al., 2018) | | 137 862 321 8 | 52560 17544 26304 7588 | 10 min 1 hour 1 day | 2016 2015 2012 1990 | RSE CORR | 6/2/2 |

Table 2: Multivariate forecasting performance of IRN and baseline models on the ETT datasets. Best results are highlighted in bold.

| Model | Metrics | ETTh1 | | | | | ETTh2 | | | | | ETTm1 | | | | |
|---|---|---|---|---|---|---|---|---|---|---|---|---|---|---|---|---|
| | | 24 | 48 | 168 | 336 | 720 | 24 | 48 | 168 | 336 | 720 | 24 | 48 | 96 | 288 | 672 |
| LogTrans | MSE | 0.686 | 0.766 | 1.002 | 1.362 | 1.397 | 0.828 | 1.806 | 4.07 | 3.875 | 3.913 | 0.419 | 0.507 | 0.768 | 1.462 | 1.669 |
| | MAE | 0.604 | 0.757 | 0.846 | 0.952 | 1.291 | 0.75 | 1.034 | 1.681 | 1.763 | 1.552 | 0.412 | 0.583 | 0.792 | 1.32 | 1.461 |
| Reformer | MSE | 0.991 | 1.313 | 1.824 | 2.117 | 2.415 | 1.531 | 1.871 | 4.66 | 4.028 | 5.381 | 0.724 | 1.098 | 1.433 | 1.82 | 2.187 |
| | MAE | 0.754 | 0.906 | 1.138 | 1.28 | 1.52 | 1.613 | 1.735 | 1.846 | 1.688 | 2.015 | 0.607 | 0.777 | 0.945 | 1.094 | 1.232 |
| TCC | MSE | 0.766 | 0.825 | 0.982 | 1.099 | 1.267 | 1.154 | 1.579 | 3.456 | 3.184 | 3.538 | 0.502 | 0.645 | 0.675 | 0.758 | 0.854 |
| | MAE | 0.629 | 0.657 | 0.731 | 0.786 | 0.859 | 0.838 | 0.983 | 1.459 | 1.42 | 1.523 | 0.478 | 0.559 | 0.583 | 0.633 | 0.689 |
| TST | MSE | 0.735 | 0.8 | 0.973 | 1.029 | 1.02 | 0.994 | 1.159 | 2.609 | 2.824 | 2.684 | 0.471 | 0.614 | 0.645 | 0.749 | 0.857 |
| | MAE | 0.633 | 0.671 | 0.768 | 0.797 | 0.798 | 0.779 | 0.85 | 1.265 | 1.337 | 1.334 | 0.491 | 0.56 | 0.581 | 0.644 | 0.709 |
| CPC | MSE | 0.728 | 0.774 | 0.92 | 1.05 | 1.16 | 0.551 | 0.752 | 2.452 | 2.664 | 2.863 | 0.478 | 0.641 | 0.707 | 0.781 | 0.88 |
| | MAE | 0.6 | 0.629 | 0.714 | 0.779 | 0.835 | 0.572 | 0.684 | 1.213 | 1.304 | 1.399 | 0.459 | 0.55 | 0.593 | 0.644 | 0.7 |
| Triplet | MSE | 0.942 | 0.975 | 1.135 | 1.187 | 1.283 | 1.285 | 1.455 | 2.175 | 2.007 | 2.157 | 0.689 | 0.752 | 0.744 | 0.808 | 0.917 |
| | MAE | 0.729 | 0.746 | 0.825 | 0.859 | 0.916 | 0.911 | 0.966 | 1.155 | 1.101 | 1.139 | 0.592 | 0.624 | 0.623 | 0.662 | 0.72 |
| MoCo | MSE | 0.623 | 0.669 | 0.82 | 0.981 | 1.138 | 0.444 | 0.613 | 1.791 | 2.241 | 2.425 | 0.458 | 0.594 | 0.621 | 0.7 | 0.821 |
| | MAE | 0.555 | 0.586 | 0.674 | 0.755 | 0.831 | 0.495 | 0.595 | 1.034 | 1.186 | 1.292 | 0.444 | 0.528 | 0.553 | 0.606 | 0.674 |
| TNC | MSE | 0.708 | 0.749 | 0.884 | 1.02 | 1.157 | 0.612 | 0.84 | 2.359 | 2.782 | 2.753 | 0.522 | 0.695 | 0.731 | 0.818 | 0.932 |
| | MAE | 0.592 | 0.619 | 0.699 | 0.768 | 0.83 | 0.592 | 0.716 | 1.213 | 1.349 | 1.394 | 0.472 | 0.567 | 0.595 | 0.649 | 0.712 |
| Informer | MSE | 0.577 | 0.685 | 0.931 | 1.128 | 1.215 | 0.72 | 1.457 | 3.489 | 2.723 | 3.467 | 0.323 | 0.494 | 0.678 | 1.056 | 1.192 |
| | MAE | 0.549 | 0.625 | 0.752 | 0.873 | 0.896 | 0.665 | 1.001 | 1.515 | 1.34 | 1.473 | 0.369 | 0.503 | 0.614 | 0.786 | 0.926 |
| TS2Vec | MSE | 0.59 | 0.624 | 0.762 | 0.931 | 1.063 | 0.423 | 0.619 | 1.845 | 2.194 | 2.636 | 0.453 | 0.592 | 0.635 | 0.693 | 0.782 |
| | MAE | 0.531 | 0.555 | 0.639 | 0.728 | 0.799 | 0.489 | 0.605 | 1.074 | 1.197 | 1.37 | 0.444 | 0.521 | 0.554 | 0.597 | 0.653 |
| SCInet | MSE | 0.341 | 0.368 | 0.451 | 0.502 | 0.583 | 0.188 | 0.279 | 0.505 | 0.618 | 1.074 | 0.126 | 0.169 | 0.191 | 0.365 | 0.713 |
| | MAE | 0.379 | 0.395 | 0.457 | 0.497 | 0.56 | 0.288 | 0.358 | 0.504 | 0.56 | 0.761 | 0.229 | 0.274 | 0.291 | 0.415 | 0.604 |
| Pyraformer | MSE | - | - | 0.808 | 0.945 | 1.022 | - | - | - | - | - | - | - | 0.48 | 0.754 | 0.857 |
| | MAE | - | - | 0.683 | 0.766 | 0.806 | - | - | - | - | - | - | - | 0.486 | 0.659 | 0.707 |
| Cost | MSE | 0.386 | 0.437 | 0.643 | 0.812 | 0.97 | 0.447 | 0.699 | 1.549 | 1.749 | 1.971 | 0.246 | 0.331 | 0.378 | 0.472 | 0.62 |
| | MAE | 0.379 | 0.464 | 0.582 | 0.679 | 0.771 | 0.502 | 0.637 | 0.982 | 1.042 | 1.092 | 0.329 | 0.386 | 0.419 | 0.486 | 0.574 |
| Autoformer | MSE | 0.384 | 0.392 | 0.49 | 0.505 | 0.498 | 0.261 | 0.312 | 0.457 | **0.471** | **0.474** | 0.383 | 0.454 | 0.481 | 0.634 | 0.606 |
| | MAE | 0.425 | 0.419 | 0.481 | 0.484 | **0.5** | 0.341 | 0.373 | 0.455 | 0.475 | 0.484 | 0.403 | 0.453 | 0.463 | 0.528 | 0.542 |
| IRN | MSE | **0.314** | **0.343** | **0.429** | **0.467** | **0.49** | **0.182** | **0.241** | **0.437** | 0.51 | 1.07 | **0.124** | **0.143** | **0.184** | **0.342** | **0.559** |
| | MAE | **0.361** | **0.368** | **0.432** | **0.474** | 0.501 | **0.27** | **0.314** | **0.453** | 0.498 | 0.745 | **0.223** | **0.249** | **0.28** | **0.398** | **0.522** |

# 4 EXPERIMENTS

We conduct experiments on 11 real-world time series datasets and compare the performance with the latest baselines. We analyze the circumstances in which proposed IRB improves the forecasting performance. We refer base model (Liu et al., 2021a) for the experiment settings. Due to page limits, the implementation details including the loss function, datasets, and metrics are reported in the Appendix.

## 4.1 DATASET

Experiments are conducted on following time series datasets: Electricity Transformer Temperature (Zhou et al., 2021), PEMS (Chen et al., 2001), Solar, Traffic, Electricity, Exchange-rate (Lai et al., 2018). The datasets, experiment settings, and metrics are summarized in Table 1.

Table 3: Univariate forecasting performance of IRN and baseline models on the ETT datasets. Best results are highlighted in bold.

| Model | Metrics | ETTh1 | | | | | ETTh2 | | | | | ETTm1 | | | | |
|---|---|---|---|---|---|---|---|---|---|---|---|---|---|---|---|---|
| | | 24 | 48 | 168 | 336 | 720 | 24 | 48 | 168 | 336 | 720 | 24 | 48 | 96 | 288 | 672 |
| N-BEAT | MSE | 0.042 | 0.065 | 0.106 | 0.127 | 0.269 | 0.078 | 0.123 | 0.244 | 0.27 | 0.281 | 0.031 | 0.056 | 0.095 | 0.157 | 0.207 |
| | MAE | 0.156 | 0.2 | 0.255 | 0.284 | 0.422 | 0.21 | 0.271 | 0.393 | 0.418 | 0.432 | 0.117 | 0.168 | 0.234 | 0.311 | 0.37 |
| Informer | MSE | 0.098 | 0.158 | 0.183 | 0.222 | 0.269 | 0.093 | 0.155 | 0.232 | 0.263 | 0.277 | 0.03 | 0.069 | 0.194 | 0.401 | 0.512 |
| | MAE | 0.247 | 0.319 | 0.346 | 0.387 | 0.435 | 0.24 | 0.314 | 0.389 | 0.417 | 0.431 | 0.137 | 0.203 | 0.372 | 0.554 | 0.644 |
| TCC | MSE | 0.053 | 0.074 | 0.133 | 0.161 | 0.176 | 0.111 | 0.148 | 0.225 | 0.232 | 0.242 | 0.026 | 0.045 | 0.072 | 0.158 | 0.239 |
| | MAE | 0.175 | 0.209 | 0.284 | 0.32 | 0.343 | 0.255 | 0.298 | 0.374 | 0.385 | 0.397 | 0.122 | 0.165 | 0.211 | 0.318 | 0.398 |
| TST | MSE | 0.127 | 0.202 | 0.491 | 0.526 | 0.717 | 0.134 | 0.171 | 0.261 | 0.269 | 0.278 | 0.048 | 0.064 | 0.102 | 0.172 | 0.224 |
| | MAE | 0.284 | 0.362 | 0.596 | 0.618 | 0.76 | 0.281 | 0.321 | 0.404 | 0.413 | 0.42 | 0.151 | 0.183 | 0.231 | 0.316 | 0.366 |
| CPC | MSE | 0.076 | 0.104 | 0.162 | 0.183 | 0.212 | 0.109 | 0.152 | 0.251 | 0.238 | 0.234 | 0.018 | 0.035 | 0.059 | 0.118 | 0.177 |
| | MAE | 0.217 | 0.259 | 0.326 | 0.351 | 0.387 | 0.251 | 0.301 | 0.392 | 0.388 | 0.389 | 0.035 | 0.142 | 0.188 | 0.271 | 0.332 |
| Triplet | MSE | 0.13 | 0.145 | 0.173 | 0.167 | 0.195 | 0.16 | 0.181 | 0.214 | 0.232 | 0.251 | 0.071 | 0.084 | 0.097 | 0.13 | 0.16 |
| | MAE | 0.289 | 0.306 | 0.336 | 0.333 | 0.368 | 0.316 | 0.339 | 0.372 | 0.389 | 0.406 | 0.18 | 0.206 | 0.23 | 0.276 | 0.315 |
| MoCo | MSE | 0.04 | 0.063 | 0.122 | 0.144 | 0.183 | 0.095 | 0.13 | 0.204 | 0.206 | **0.206** | 0.015 | 0.027 | 0.041 | 0.083 | 0.122 |
| | MAE | 0.151 | 0.191 | 0.268 | 0.297 | 0.347 | 0.234 | 0.279 | 0.36 | 0.364 | **0.369** | 0.091 | 0.122 | 0.153 | 0.219 | 0.268 |
| TNC | MSE | 0.057 | 0.094 | 0.171 | 0.192 | 0.235 | 0.097 | 0.131 | 0.197 | 0.207 | 0.207 | 0.19 | 0.036 | 0.054 | 0.098 | 0.136 |
| | MAE | 0.184 | 0.239 | 0.329 | 0.357 | 0.408 | 0.238 | 0.281 | 0.354 | 0.366 | 0.37 | 0.103 | 0.142 | 0.178 | 0.244 | 0.29 |
| TS2Vec | MSE | 0.039 | 0.062 | 0.142 | 0.16 | 0.179 | 0.091 | 0.124 | 0.198 | 0.205 | 0.208 | 0.016 | 0.028 | 0.045 | 0.095 | 0.142 |
| | MAE | 0.151 | 0.189 | 0.291 | 0.316 | 0.345 | 0.23 | 0.274 | 0.355 | 0.364 | 0.371 | 0.093 | 0.126 | 0.162 | 0.235 | 0.29 |
| SCInet | MSE | 0.031 | 0.051 | 0.081 | 0.094 | 0.176 | 0.07 | 0.102 | 0.157 | 0.177 | 0.253 | 0.019 | 0.045 | 0.072 | 0.117 | 0.18 |
| | MAE | 0.132 | 0.173 | 0.222 | 0.242 | 0.343 | 0.194 | 0.242 | 0.311 | 0.34 | 0.403 | 0.088 | 0.143 | 0.198 | 0.266 | 0.328 |
| CoST | MSE | 0.04 | 0.06 | 0.097 | 0.112 | **0.148** | 0.079 | 0.118 | 0.189 | 0.206 | 0.214 | **0.015** | **0.025** | **0.038** | **0.077** | **0.113** |
| | MAE | 0.152 | 0.186 | 0.236 | 0.258 | **0.306** | 0.207 | 0.259 | 0.339 | 0.36 | 0.371 | 0.088 | **0.117** | **0.147** | **0.209** | **0.257** |
| IRN | MSE | **0.03** | **0.045** | **0.078** | **0.091** | 0.168 | **0.067** | **0.093** | **0.154** | **0.172** | 0.235 | 0.018 | 0.043 | 0.07 | 0.116 | 0.151 |
| | MAE | **0.131** | **0.163** | **0.218** | **0.241** | 0.329 | **0.189** | **0.232** | **0.31** | **0.337** | 0.392 | **0.087** | 0.141 | 0.196 | 0.264 | 0.301 |

Table 4: Forecasting performance of IRN and baseline models on PEMS datasets. Best results are highlighted in bold.

| Model | PEMS03 | | | PEMS04 | | | PEMS07 | | | PEMS08 | | |
|---|---|---|---|---|---|---|---|---|---|---|---|---|
| | MAE | RMSE | MAPE (%) | MAE | RMSE | MAPE (%) | MAE | RMSE | MAPE (%) | MAE | RMSE | MAPE (%) |
| LSTM | 21.33 | 35.11 | 21.33 | 25.14 | 39.59 | 20.33 | 29.98 | 42.84 | 15.33 | 22.2 | 32.06 | 15.32 |
| TCN | 18.87 | 32.24 | 18.63 | 22.81 | 36.87 | 14.31 | 30.53 | 41.02 | 13.88 | 21.42 | 34.03 | 13.09 |
| DCRNN | 18.18 | 30.31 | 18.91 | 24.7 | 38.12 | 17.12 | 28.3 | 38.58 | 11.66 | 17.86 | 27.83 | 11.45 |
| STGCN | 17.49 | 30.12 | 17.15 | 22.7 | 35.55 | 14.59 | 25.38 | 38.78 | 11.08 | 18.02 | 27.83 | 11.4 |
| ASTGCN(r) | 17.69 | 29.66 | 19.4 | 22.93 | 35.22 | 16.56 | 28.05 | 42.57 | 13.92 | 18.61 | 28.16 | 13.08 |
| STSGCN | 17.48 | 29.21 | 16.78 | 21.19 | 33.65 | 13.9 | 24.26 | 39.03 | 10.21 | 19.13 | 31.05 | 12.68 |
| STFGNN | 16.77 | 26.28 | 16.3 | 20.48 | 32.51 | 16.77 | 23.46 | 36.6 | 9.21 | 17.13 | 26.8 | 10.96 |
| AGCRN | 15.98 | 28.25 | 15.23 | 19.83 | 32.3 | 12.97 | 22.37 | 36.55 | 9.12 | 16.94 | 26.25 | 10.6 |
| SCInet | 15 | 24.31 | 14.29 | **18.95** | 30.89 | 11.86 | 21.19 | 34.03 | **8.83** | 15.72 | 24.76 | **9.8** |
| DSTAGNN | 15.57 | 27.21 | 14.68 | 19.3 | 31.46 | 12.7 | 21.42 | 34.51 | 9.01 | 15.9 | 25.24 | 9.97 |
| IRN | **14.98** | **23.99** | **14.18** | 19.03 | **30.88** | **11.71** | **21.11** | **33.99** | 8.84 | **15.71** | **24.64** | **9.8** |

## 4.2 BASELINES

For each dataset, we compare IRN with the latest baselines: (1) For **ETT**, Transformer-based methods (i.e., LogTrans (Li et al., 2019), Informer (Zhou et al., 2021) Autoformer (Wu et al., 2021), Reformer (Kitaev et al., 2020), TST (Zerveas et al., 2021), and Pyraformer (Liu et al., 2021b)) and feature representation learning based methods (i.e., TCC (Eldele et al., 2021), N-BEATS (Oreshkin et al., 2019), CPC (Oord et al., 2018), Triplet (Franceschi et al., 2019), MoCo (He et al., 2020), TNC (Tonekaboni et al., 2021), TS2Vec (Yue et al., 2022), SCInet(Liu et al., 2021a) and CoST (Woo et al., 2022) ); (2) For **PEMS**, LSTM (Hochreiter & Schmidhuber, 1997), CNN-based methods (i.e., TCN and DCRNN (Li et al., 2017)), SCInet, Graph-based methods (i.e., STGCN (Yu et al., 2017), AST-GCNr (Guo et al., 2019), STSGCN (Song et al., 2020), STFGNN (Li & Zhu, 2021), AGCRN (Bai et al., 2020), and DSTAGNN (Lan et al., 2022)); (3) For **Solar Energy, Traffic, Electricity, and Exchange Rate**, AR, VAR-MLP (Zhang, 2003), GP (Frigola, 2015), GRU, LSTNet (Lai et al., 2018), TPA-LSTM (Shih et al., 2019), SCInet, and MTGNN (Wu et al., 2020).

## 4.3 EXPERIMENTAL RESULTS

We summarize the performances of IRN and baseline models in Table 2 to 5. IRN demonstrates state-of-the-art performances in 36 cases and near-best in 14 cases. Autoformer performs better for long-term forecasting in ETTh2 datasets as it shows strengths in reflecting trends and seasonality, which are more apparent in longer sequences (Wu et al., 2021). In a similar vein, features are more

Table 5: Forecasting performance comparison of IRN and baseline models on the Solar-Energy, Traffic, Electricity, and Exchange-rate datasets. Best results are highlighted in bold.

| Methods | Metrics | Solar-Energy | | | | Traffic | | | | Electricity | | | | Exchange-rate | | | |
|---|---|---|---|---|---|---|---|---|---|---|---|---|---|---|---|---|---|
| | | 3 | 6 | 12 | 24 | 3 | 6 | 12 | 24 | 3 | 6 | 12 | 24 | 3 | 6 | 12 | 24 |
| AR | RSE | 0.2435 | 0.379 | 0.5911 | 0.8699 | 0.5991 | 0.6218 | 0.6252 | 0.63 | 0.0995 | 0.1035 | 0.105 | 0.1054 | 0.0228 | 0.0279 | 0.0353 | 0.0445 |
| | CORR | 0.971 | 0.9263 | 0.8107 | 0.5314 | 0.7752 | 0.7568 | 0.7544 | 0.7591 | 0.8845 | 0.8632 | 0.8691 | 0.8595 | 0.9734 | 0.9656 | 0.9526 | 0.9357 |
| VARMLP | RSE | 0.1922 | 0.2679 | 0.4244 | 0.6841 | 0.5582 | 0.6579 | 0.6023 | 0.6146 | 0.1393 | 0.162 | 0.1557 | 0.1274 | 0.0265 | 0.0394 | 0.0407 | 0.0578 |
| | CORR | 0.9829 | 0.9655 | 0.9058 | 0.7149 | 0.8245 | 0.7695 | 0.7929 | 0.7891 | 0.8708 | 0.8389 | 0.8192 | 0.8679 | 0.8609 | 0.8725 | 0.828 | 0.7675 |
| GP | RSE | 0.2259 | 0.3286 | 0.52 | 0.7973 | 0.6082 | 0.6772 | 0.6406 | 5995 | 0.15 | 0.1907 | 0.1621 | 0.1273 | 0.0239 | 0.0272 | 0.0394 | 0.058 |
| | CORR | 0.9751 | 0.9448 | 0.8518 | 0.5971 | 0.7831 | 0.7406 | 0.7671 | 0.7909 | 0.867 | 0.8334 | 0.8394 | 0.8818 | 0.8713 | 0.8193 | 0.8484 | 0.8278 |
| RNN_GRU | RSE | 0.1932 | 0.2628 | 0.4163 | 0.4852 | 0.5358 | 0.5522 | 0.5562 | 5633 | 0.1102 | 0.1144 | 0.1183 | 0.1295 | 0.0192 | 0.0264 | 0.0408 | 0.0626 |
| | CORR | 0.9823 | 0.9675 | 0.915 | 0.8823 | 0.8511 | 0.8405 | 0.8345 | 0.83 | 0.8597 | 0.8623 | 0.8472 | 0.8651 | 0.9786 | 0.9712 | 0.9531 | 0.9223 |
| LSTNet | RSE | 0.1843 | 0.2559 | 0.3254 | 0.4643 | 0.4777 | 0.4893 | 0.495 | 0.4973 | 0.0864 | 0.0931 | 0.1007 | 0.1007 | 0.0226 | 0.028 | 0.0356 | 0.0449 |
| | CORR | 0.9843 | 0.969 | 0.9467 | 0.887 | 0.8721 | 0.869 | 0.8614 | 0.8588 | 0.9283 | 0.9135 | 0.9077 | 0.9119 | 0.9735 | 0.9658 | 0.9511 | 0.9354 |
| SCInet | RSE | 0.1775 | 0.2301 | 0.2997 | 0.4081 | 0.4216 | 0.4414 | 0.4495 | 0.4453 | 0.0748 | 0.0845 | 0.0926 | 0.0976 | 0.018 | 0.0247 | 0.034 | 0.0442 |
| | CORR | 0.9853 | 0.9739 | 0.955 | 0.9112 | 0.892 | 0.8809 | 0.8772 | 0.8825 | 0.9492 | 0.9386 | 0.9304 | 0.9274 | 0.9739 | 0.9662 | 0.9487 | 0.9255 |
| TPA-LSTM | RSE | 0.1803 | 0.2347 | 0.3234 | 0.4389 | 0.4487 | 0.4658 | 0.4641 | 0.4765 | 0.0823 | 0.0916 | 0.0964 | 0.1006 | **0.174** | **0.241** | 0.0341 | 0.0444 |
| | CORR | 0.985 | 0.9742 | 0.9487 | 0.9081 | 0.8812 | 0.8717 | 0.8794 | 0.8629 | 0.9439 | 0.9337 | 0.925 | 0.9133 | **0.979** | **0.9709** | **0.9564** | **0.9381** |
| MTGNN | RSE | 0.1778 | 0.2348 | 0.3109 | 0.427 | **0.4162** | 0.4754 | **0.4461** | 0.4535 | 0.0745 | 0.0878 | **0.0916** | **0.0953** | 0.0194 | 0.0259 | 0.0349 | 0.0456 |
| | CORR | 0.9852 | 0.9726 | 0.9509 | 0.9031 | **0.8963** | 0.8667 | **0.8794** | 0.881 | 0.9474 | 0.9316 | 0.9278 | 0.9234 | 0.9786 | 0.9708 | 0.9551 | 0.9372 |
| IRN | RSE | **0.1770** | **0.2292** | **0.2971** | **0.4050** | 0.4171 | **0.4349** | 0.4493 | **0.4449** | **0.0739** | **0.0844** | 0.0926 | 0.0968 | **0.0179** | **0.0246** | **0.0337** | **0.0441** |
| | CORR | **0.9853** | **0.9853** | **0.9556** | **0.9112** | 0.8920 | **0.8861** | 0.8774 | **0.8788** | **0.9493** | **0.939** | **0.9313** | **0.9281** | 0.9765 | 0.9678 | 0.9522 | 0.9288 |

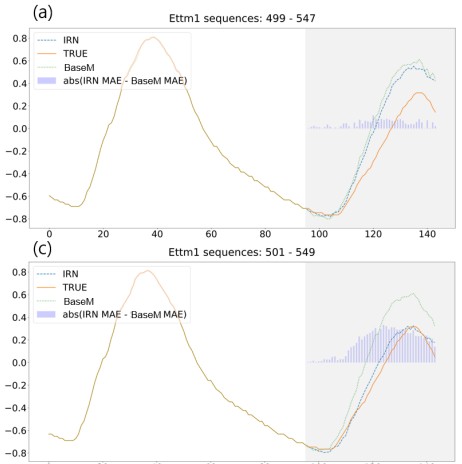
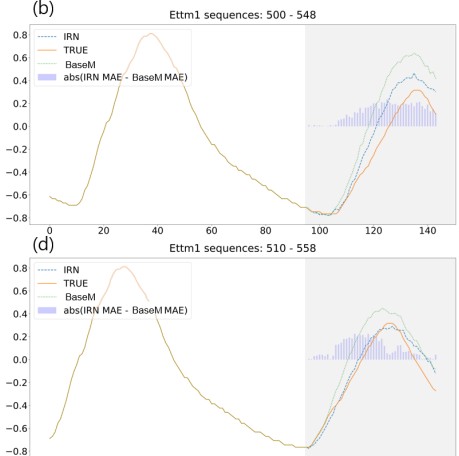

Figure 5: Forecasting results of IRN and the base model from (a) sequence 499 to 547, (b) sequence 500 to 548, (c) sequence 501 to 549, (d) sequence 510 to 558 in Ettm1 data. The ground truths are shown in a solid line. The dotted and dashed lines represent the predicted values of the base model and IRN, respectively. The predicting region is shaded in grey. The bar graph shows the absolute MAE difference between the base model and IRN.

evident in univariate time series, which explains the higher performances of MoCo (He et al., 2020) and CoST (Woo et al., 2022), which are feature representation learning models, on 6 cases of ETT univariate datasets. MTGNN (Wu et al., 2020), a model specialized for analyzing edge relations, yields the best performance on Traffic and Electricity datasets. This is because both datasets contain complex edge between nodes. At last, compared to the attention-based model, IRN shows lesser performances on Exchange-rate dataset due to the strong random-walk property of the time series (Wright, 2008). Overall, our IRN successfully reflects irregularity representation and complements base model to achieve the higher forecasting performances.

## 4.4 ABLATION STUDY

We perform the ablation study to demonstrate the benefit obtained by IRB. We plot the ground truths and corresponding predictions of IRN and the base model at 499th, 500th, 501th, and 510th sequences of ETTh1 data as shown Figure 5. In Figure 5 (a), the original time series has a peak in the predicting region shaded in grey. Up to sequence 499, IRN and the base model make similar predictions having large errors. When a sequence is added as in the Figure 5 (b), the discrepancy between the ground truth and the predicted values of IRN decreases. With an additional sequence in the Figure 5 (c), IRN quickly reflects the change and makes a better forecast than the base model. It is observed that the base model is less sensitive to the change of the input sequences, giving similar

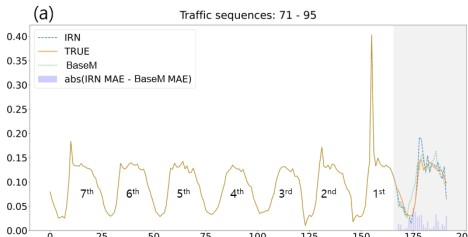 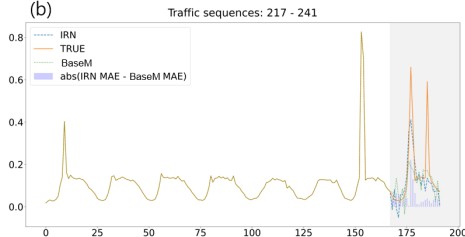

Figure 6: Forecasting results of (a) sequence 71 to 95 and (b) sequence 217 to 241 in traffic data using IRN and base model.

predictions from sequence 499 to sequence 501. Only when 11 sequences have passed, the base model considers the actual changes as in Figure 5 (d). We verify that IRN can reflect the irregular features (instantaneous changes).

Next, we observe the 24 horizon forecasts in Traffic dataset for further analysis. In Figure 6 (a), cycles 2 to 7 consist of values lower than 0.2, whereas cycle 1 includes irregular values greater than 0.3. IRN has larger errors than the base model as IRN instantaneously reflects the irregularity. In contrast, IRN performs better than the base model when the irregularity persists as shown in Figure 6 (b). The reflection of the irregularity does not always end in a better forecast, but IRB consistently improves the forecasting performance of the base model, which confirms the effectiveness of the our model.

## 4.5 Observation on the variation of the irregularity

We further investigate how the performance of IRN changes with the irregularity of the time series. Ettm1 and Traffic datasets are decomposed into seasonality, trend, and irregularity as depicted in Figure 1. The variation of the irregularity is calculated and classified into two cases according to the degree of variation. Case 1 and Case 2 represent 500 data points with low variation of irregularity and high variation of irregularity, respectively. The difference of average Mean Square Error

Table 6: The difference of average MSE between IRN and base model according to the variation of irregularity on ETTm1 and Traffic datasets. Case 1 and Case 2 refer to 500 data points with low variation in irregularity and with high variation in irregularity, respectively.

| Dataset | Horizon | MSE difference between IRN and base model | |
| | | Case 1 | Case 2 |
|---|---|---|---|
| Ettm1 | 48 | -0.00097 | 0.001506 |
| Traffic | 24 | 0.00709 | 0.00941 |

(MSE) between IRN and base model are obtained for each case and listed in Table 6. This results indicate that higher performance improvement is attained in case 2 than case 1 for both datasets, implying the higher the irregularity variation, the higher performance improvement can be achieved.

## 5 Conclusion

In this paper, we propose Irregularity Reflection Neural Network (IRN), a deep learning based model for time series forecasting that reflects the irregularity in time series. We introduce a novel expression of irregularity based on Fourier series under microscopic perspective condition and employ it to design the Irregularity Representation Block (IRB) that captures, preserves, and learns the irregularity representation of time series data. By embedding the IRB on the base model, IRN is further proposed. Experiments on a variety of real-world datasets show that IRN can consistently outperform existing state-of-the-art baselines. The ablation study confirm that the proposed methodology can reflect the irregularity. Accordingly, we argue that the irregularity representation is essential for improving performance of machine learning models.

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

## A  LOSS FUNCTION

We cover stacked cases in which losses are accumulated. When the dataset has enough training data, we apply K layers (Liu et al., 2021a). To train the K stacked IRN for the k-th intermediate prediction, we compute the L1 loss between the k-th prediction and the true value as follows:

$$L_k = \frac{1}{h} \sum_{i=0}^{h} |\hat{y}_i^k - y_i|$$

where $h$ is the horizon size, $k$ is the number of stacks, $\hat{y}^k$ is $i$-th horizon prediction of k-th stack, and $y$ is the true value. We apply Equation 1 to calculate the L1 loss of each stacked layer output. The total loss of the stacked IRN is expressed as:

$$L = \sum_{k=0}^{K} L_k$$

## B    IMPLEMENTATION DETAILS

Our model and framework are implemented with Pytorch. We train IRN with Adam optimizer by using NVIDIA 2080Ti 8 GPUs for enough batch size. Other parameters such as learning rate, level, stack, single, and multi are changed according to the dataset charateristics and referring base model (Liu et al., 2021a).

## C    DATASETS AND METRICS

### C.1    ELECTRICITY TRANSFORMER TEMPERATURE

ETT contains two-year electric power data gathered from two counties in China (hourly subsets ETTh1, ETTh2 and 15 minutes subsets ETTm1). Each data point contains an oil temperature value and six power load components. The train, validation and test sets consist of 12, 4, and 4 months data, respectively. We implement zero-mean normalization for data pre-processing. Mean Absolute Errors (MAE) (Hyndman & Koehler, 2006) and Mean Squared Errors (MSE) (Makridakis et al., 1982) are used as evaluation metrics.

$$MAE = \frac{1}{h} \sum_{i=0}^{h} |\hat{x}_i - x_i|$$

$$MSE = \frac{1}{h} \sum_{i=0}^{h} (\hat{x}_i - x_i)^2$$

where $x_i$ is the true value, $\hat{x}_i$ is the predicted value, and h is the prediction horizon size.

### C.2    PEMS

PeMS consists of four public datasets (i.e., PEMS03, PEMS04, PEMS07 and PEMS08), which are separately collected from Caltrans Performance Measurement System (PeMS) of four sections in California. The data is collected every five minutes. We predict one hour that consists of 12 data points. The zero-mean normalization is applied for the data pre-processing. The evaluation metrics are MAE, Root Mean Squared Errors (RMSE) and Mean Absolute Percentage Errors (MAPE).

$$RMSE = \sqrt{\frac{1}{h} \sum_{i=0}^{h} (\hat{x}_i - x_i)^2}$$

$$MAPE = \frac{1}{h} \sum_{i=0}^{h} |\frac{(\hat{x}_i - x_i)}{x_i}|$$

### C.3    TRAFFIC, SOLAR ENERGY, ELECTRICITY AND EXCHANGE RATE

Traffic includes the hourly road occupancy rates which ranges from 0 to 1. The sensors gather the road occupancy rates from 2015 to 2016. Solar Energy contains 2016 solar power production which are recorded every 10 minutes from 137 PV plants in Alabama State. Electricity collects the hourly electricity consumption (kWh) of 321 clients from 2012 to 2014. Exchange-Rate consists of the daily exchange rates of 8 foreign countries from 1990 to 2016. For four datasets, the size of the lookback window is 168, and horizon sizes are 3,6,12, and 24. The evaluation metrics are Root Relative Squared Error (RSE) and Empirical Correlation Coefficient (CORR) (Lai et al., 2018).

$$RSE = \frac{\sqrt{\sum_{i=0}^{h} (\hat{x}_i - x_i)^2}}{\sqrt{\sum_{i=0}^{h} (x_i - mean(x))^2}}$$

$$CORR = \frac{1}{d}\sum_{j=0}^{d} \frac{\sum_{i=0}^{h}(x_{i,j} - mean(x_j))(\hat{x}_{i,j} - mean(\hat{x}_j)}{\sum_{i=0}^{h}(x_{i,j} - mean(x_j))^2(\hat{x}_{i,j} - mean(\hat{x}_j))^2}$$

where $d$ is the number of variates.

