# OpenReview forum: "Irregularity Reflection Neural Network for Time Series Forecasting"
_ICLR.cc/2023/Conference — Submitted to ICLR 2023_

### Official Review · Reviewer_aUx9 · 2022-10-24

**Confidence:** 3
**Clarity, Quality, Novelty And Reproducibility:** The paper is well written overall
**Correctness:** 3
**Technical Novelty And Significance:** 3
**Empirical Novelty And Significance:** 3
**Recommendation:** 6

**Strength And Weaknesses:**

Mathematical models are used to explain the methodology and extensive experiments on a variety of real word datasets are provided.

The code of the work is not available making difficult to evaluate if the result can be reproduced easily


**Summary Of The Paper:**

The paper proposes a deep learning model for time series forecasting that reflects the irregularity in time series. Irregularity is based on Fourier series and and the authors employ it to design the Irregularity Representation Block  that captures, preserves, and learns the irregularity representation of time series data. Experiments on a variety of realworld datasets show the efficiency of the method.



**Summary Of The Review:**

The paper is interesting overall with mathematical models helping to understand the main concepts. Several experiments are provided to show the efficiency of the method, but we do not have access to the code.

---

### Official Review · Reviewer_ppXw · 2022-10-25

**Confidence:** 4
**Clarity, Quality, Novelty And Reproducibility:** See Summary of the review
**Correctness:** 4
**Technical Novelty And Significance:** 3
**Empirical Novelty And Significance:** 3
**Recommendation:** 6

**Strength And Weaknesses:**

Strength:
1. Residual modeling of time-series data is quite under explored which gives the research direction of this literature great value.
2. Authors performed extensive experiments and ablation studies on number of real-world dataset to prove the validity of their approach

Weakness:
1. The authors didn’t provide any link to their code for reproducibility purpose, so it won’t be easy for peers to validate the experimental results
2. The technical depth of the work is a tad limited

**Summary Of The Paper:**

Summary: While time-series components like trends or seasonality have been extensively studied and been modeled in existing literature,
 	modeling residual or irregularity has been an under explored research area due to its inherent complexity.
	In this work, the authors propose a deep learning based model for time-series forecasting (TSF) that reflects this irregularity
 	by introducing a novel expression for residual and employing it in TSF.

**Summary Of The Review:**

Short review:
1. The paper’s presentation is decent and comprehensible
2. This work focuses on an under explored research area and shows great result when applied in real-world dataset
3. The paper doesn’t show the best technical depth
4. No link to the code available

---

> ### Author Response · Authors · 2022-11-11
> **Thank you for your valuable review**
>
> We express our sincere thanks for taking the time to review our paper.
>
> Sorry for the inconvenience made regarding the code. We left a comment in common for the code execution.
>
> If there is any issues in executing the code, please let me know and I will reply as soon as possible. :)

---

### Official Review · Reviewer_ZKbd · 2022-10-25

**Confidence:** 4
**Correctness:** 2
**Technical Novelty And Significance:** 2
**Empirical Novelty And Significance:** 2
**Recommendation:** 3

**Clarity, Quality, Novelty And Reproducibility:**

The paper is mostly well written, but sufferes from
a. unexplained symbols in the formulas (e.g. "R_CAV" in eq. 7,
  you only explain "R"; what is "t_{a~b}" ?),
b. many typos, e.g.,
  p. 2 "models is"
  p. 2 "the increasing the number"

Smaller points:
- tab. 3 and 4 are hard to read and miss some information:
  - mixing MSE and MAE rows makes not so much sense, as you never want to compare
    them.
  - standard deviations would make it easier to assess if we see significant differences
    or noise.

The proposed method combines standard components like convolutions
and pooling layers, but the specific cell that is constructed from those
components is novel.

The reproducibility is medium (experiments explained clearly, but no code
provided).


**Strength And Weaknesses:**

s1. very simple method.
s2. promising experimental results.

w1. misses a closely related, even more simple work.
w2. how the proposed model mimicks the Fourier transform is not clear.
w3. the concept of irregularity and the derivation of the approximation
  is unclear.


**Summary Of The Paper:**

The paper addresses the problem of time series forecasting.
The authors propose a model that combines convolutions and
min, max and average poolings, aiming to mimick a first order
approximation of a Fourier transformation. In experiments
they show that their method outperforms existing baselines.


**Summary Of The Review:**

I see three issues with the paper:

w1. misses a closely related, even more simple work.

  The DLinear model by Zeng et al. 2022 proposes an even simpler model with
  comparable results to yours. What advantages does your model have?

w2. how the proposed model mimicks the Fourier transform is not clear.

  In eq. 9, you define a layer conv-relu-conv-tanh and claim that it approximates
  the sine basis function. But there is nothing in your layer that would be geared
  specifically to model a sine function, you could claim that it approximates any
  function. To me it looks like your derivation breaks here.

w3. the concept of irregularity and the derivation of the approximation
  is unclear.

  You claim that there is a third component in time series analysis besides
  seasonality and trend, "irregularity". To me this looks wrong and seems to
  misunderstand what a trend usually models in time series: often it is just
  a linear trend, and then you are perfectly right, there is room for further
  basis functions. But generally, trends can be any non-periodic basis functions,
  e.g. exponentials or logarithms etc. Between periodic ("seasonality") and
  non-periodic ("trend") basis functions I do not see a third type of basis
  functions.

  The basis function you model in the end (eq. 5) actually **is** a simple
  seasonality, a single sine function.

  The derivation of eq. 5 is problematic: the higher order terms do not
  "correspond to the vertical shift", because they also are periodic. If you
  want to make an approximation argument here, that **locally** the
  higher order terms can be approximated by a constant, you need to be
  more specific, esp. you need to say explicitly that n-th regular waves
  are the waves with **slower** angular velocity (as your fig. 3a suggests, but
  you never say that). However, in Fourier analysis by default basis functions
  are order by **increasing** angular velocity for good reasons, because
  only if you go to arbitrary high velocities you have arbitrary good
  approximations.

  So both, your concept of irregularity and the derivation of your layer
  function look problematic to me.


references:
  Ailing Zeng, Muxi Chen, Lei Zhang, Qiang Xu:
  Are Transformers Effective for Time Series Forecasting? CoRR abs/2205.13504 (2022)

---

> ### Author Response · Authors · 2022-11-11
> **Thank you for your valuable review (1)**
>
> We express our sincere thanks for taking the time to review our paper.
>
> **w0** The paper is mostly well written, but sufferes from a. unexplained symbols in the formulas (e.g. "R_CAV" in eq. 7, you only explain "R"; what is "t_{a~b}" ?), b. many typos, e.g., p. 2 "models is" p. 2 "the increasing the number"
>
> - *Answer*: Thank you for your comments, and we are sorry for the inconvenience. $R_{Cav}$ in eq.7 means the convolution layer with relu activation to estimate average values. $t_{a\sim b}$  is the time value under microscopic perspective condition. We also have provided additional details regarding this symbol in the manuscript. In addition, we made corrections regarding the typos.
>
>
> **w1**  misses a closely related, even more simple work. The DLinear model by Zeng et al. 2022 proposes an even simpler model with comparable results to yours. What advantages does your model have?
>
> - *Answer*: First, we would like to emphasize that our intention does not lie in decomposing and learning components of time series. Rather, we want to reflect the residuals, which we call "irregularity."  We focus on expressing and reflecting this irregularity, a time-independent component that is not covered by time-dependent components such as trend or seasonality. The sources of the irregularity can be random errors, influences of unknown features, or external shocks that may change the characteristics of the time series but are not captured as a trend or seasonality yet (e.g., the financial crisis of 2008 and COVID-19). We reviewed the mentioned paper. DLinear model[1] is indeed a simple model that deals with the trend and seasonality of the time series and shows good performance. However, it does not cover the irregularity that we are considering. Therefore, we believe that these two papers are not competing with each other and that one can achieve further performance improvement by incorporating our IRB with the DLinear model or other previous works that focus on the time-dependent nature of time series, as shown in our code implementation.
>
>
> **w2** how the proposed model mimicks the Fourier transform is not clear.
> In eq. 9, you define a layer conv-relu-conv-tanh and claim that it approximates the sine basis function. But there is nothing in your layer that would be geared specifically to model a sine function, you could claim that it approximates any function. To me it looks like your derivation breaks here.
>
> Equation9: $M_{phase}(x^{input}) = T_{tanh}(C_{Cphase}(R_{C_0}(x^{input}))) \approx sin(\theta (x^{input}))$
>
>
> - *Answer*: Thank you for your comments, and we want to clarify Equation 9.
> Again our goal is not to model the periodic nature of a time series but to capture irregularity using microscopic perspective conditions. That is, we view the whole time series as a sum of sine functions with very short cycles. In this sense, the comment that we *“could claim it approximates any function”* is right as we try to capture irregularity with the vertical shifts of sine functions. The choice of the sine function in the conceptualization of irregularity is one issue; there may be better conceptualization, which is a possible interesting research direction in the future. We appreciate the reviewer's comment on this issue.
> In more detail, the sine function was first defined in Equation 4 under microscopic perspective conditions. $sin(θ (t_{a\sim b}))$ cannot exceed 360 degrees, no matter how high the frequency is. Accordingly, , $sin(θ (t))$  can be transferred to $sin(θ (t_{a\sim b}))$, which is a micro-moment value between zero and 360 degrees. Therefore,  $sin(θ (t))$ can be defined as $sin(θ (t_{a\sim b}))$ function with no periodic property and expressed as a hyperbolic tangent which has the output value between 1 and -1 as depicted in Equation 9. We added microscopic perspective conditions in the sentence after Equation 9 to clarify this point.

---

> > ### Author Response · Authors · 2022-11-11
> > **Thank you for your valuable review (2)**
> >
> > **w3** the concept of irregularity and the derivation of the approximation is unclear.
> >
> > You claim that there is a third component in time series analysis besides seasonality and trend, "irregularity". To me this looks wrong and seems to misunderstand what a trend usually models in time series: often it is just a linear trend, and then you are perfectly right, there is room for further basis functions. But generally, trends can be any non-periodic basis functions, e.g. exponentials or logarithms etc. Between periodic ("seasonality") and non-periodic ("trend") basis functions I do not see a third type of basis functions.
> >
> > The basis function you model in the end (eq. 5) actually is a simple seasonality, a single sine function.
> >
> > The derivation of eq. 5 is problematic: the higher order terms do not "correspond to the vertical shift", because they also are periodic. If you want to make an approximation argument here, that locally the higher order terms can be approximated by a constant, you need to be more specific, esp. you need to say explicitly that n-th regular waves are the waves with slower angular velocity (as your fig. 3a suggests, but you never say that). However, in Fourier analysis by default basis functions are order by increasing angular velocity for good reasons, because only if you go to arbitrary high velocities you have arbitrary good approximations.
> >
> > Equation5 under microscopic perspective condition: $\psi \left ( t \right )\approx A(t)+\frac{max(t)-min(t)}{2}\times sin(\theta(t))$
> >
> >
> >
> > - *Answer*: Many econometrics studies like [3,5] and machine learning community [2,4] view general time series as
> > $ X=S+T+E$ ,
> > where X is time series data, S is seasonality component, T trend component, and E is the error component also known as residual.
> > We believe that the reason the review is asking about the third basis function for the residual stems from this decomposition. Yet, the general formalization of the residual is known to be extremely difficult. We considered both approaches, one that applies our idea of capturing irregularity after decomposition or at the time of decomposition along with trend and seasonality - *we believe this is the reviewer's understanding of our paper* - and the other before decomposition. We chose the latter because the captured and learned irregularity after the decomposition heavily depends on the results from the various decomposition mechanisms, leading to confusion about whether the resulting irregularities are from the time series itself or errors from the decomposition mechanisms. In this sense, even though the review pointed out relevant questions, we do not agree with the critiques that *"the basis function you model in the end (eq. 5) actually is a simple seasonality"* and that *"the derivation of eq. 5 is problematic … because they also are periodic."*
> > We propose a simple way to express the irregularity based on Fourier series. By viewing the time series from a microscopic perspective and referring to the Fourier series, we come up with equation 5, which suggests that irregularity can be expressed using the sum of regular waves and can be reinterpreted by average or vertical shifts. We want to clarify that captured irregularities using our approach also include trend and seasonality information, which can be modeled and trained with methods suggested in other studies.
> >
> > [1] Ailing Zeng, Muxi Chen, Lei Zhang, Qiang Xu: Are Transformers Effective for Time Series Forecasting? CoRR abs/2205.13504 (2022)
> > [2] Woo, Gerald, et al. "CoST: Contrastive Learning of Disentangled Seasonal-Trend Representations for Time Series Forecasting." International Conference on Learning Representations. 2021.
> > [3] Scott, Steven L., and Hal R. Varian. "Bayesian variable selection for nowcasting economic time series." Economic analysis of the digital economy. University of Chicago Press, 2015. 119-135
> > [4]Qiu, Jinwen, S. Rao Jammalamadaka, and Ning Ning. "Multivariate Bayesian Structural Time Series Model." J. Mach. Learn. Res. 19.1 (2018): 2744-2776.
> > [5] Cleveland, Robert B., et al. "STL: A seasonal-trend decomposition." J. Off. Stat 6.1 (1990): 3-73.

---

### Official Review · Reviewer_Lsd6 · 2022-11-26

**Confidence:** 3
**Correctness:** 3
**Technical Novelty And Significance:** 2
**Empirical Novelty And Significance:** 3
**Recommendation:** 3

**Clarity, Quality, Novelty And Reproducibility:**

I would love to see the clarity of the paper improved, so as to perform a fuller assessment of the models merits -- with details in the section above.

**Strength And Weaknesses:**

Strengths
---

Time series forecasting is a crucial problem across many application domains – with many datasets exhibiting level-trend-seasonality relationships in general. The development of models which can capture these relationships better would be very impactful for said applications.


Weaknesses
---

However, I do have several key concerns with the paper:

1. The paper does adopt some seemingly non-standard terms, which can be confusing with similar terms in general time series forecasting/representation learning literature.

     a.	Irregularity – the paper frequently describes residuals as “the irregularity”, without defining the term concretely or citing its source. Easily confused with modelling irregularly sampled time series – e.g. ICML 2022’s Modelling Irregular Time Series with Continuous Recurrent Units.

     b.	Reflection – appears to be used synonymously with “model”, and could be confused with concepts in vector transformations (https://en.wikipedia.org/wiki/Reflection_(mathematics)).

2. I am not sure I fully understand the motivations behind the architecture

     a.	Even if we were to understand “irregularity” as “residual”, the underlying model itself appears to be feeding in the inputs as-is – without any residualisation with respect to level-trend-seasonal components as implied by the summary.

     b.	The analogy of the IRN as a non-linear Fourier transform appears to breakdown when actual non-time inputs are used (i.e. x_input) – i.e. feeding a exogenous input into a sine wave would not necessarily make it a periodic basis function unless it is monotonic.

     c.	The description of the Fourier transform is also slightly unclear in equation 3

          i. Assuming $t_{a\~b}$ is a single point in time -- as implied by “When the irregularity … is observed at moment $t_{a\~b}$” – $min(t_{a\~b})$ becomes unmeaningful as it would simply equal $t_{a~b}$. If it represents a sequence, then sin(…) becomes non-meaningful.

     d. The parallels when we move into input space in equation 6 also require further explanation:

         i. Does dim(x^{input}) have to match dim(x^{irregular})? What if we only have a single input?

         ii. “Through this transformation, x^{irregular} is converted from time dependent to data dependent.”, but x^{input} does not appear to be a time index of any sort? Otherwise does dim d=1?

     e. How is the IRB used to generate multi-step forecasts in the table?

     f. How is hyperparameter optimisation performed for the model and benchmarks – details of which are crucial for reproducibility.

3. Unclear problem formulation – while the paper seemingly implies that the problem being tackled is for improving multi-horizon forecasts for datasets with strong level/trend/seasonal relationships, the precise problem is never fully defined. This makes it difficult to determine the appropriateness of the proposed benchmarks – many of which come from long-term time series forecasting (e.g. LogTran/Reformer/Pyraformer/Autoformer). Even with the simpler models – e.g. LSTM/GRU/VAR-MLP etc – how were they applied for multi-horizon forecasts?

     i. Assuming the paper is tackling models for enhanced level/trend/seasonality modelling – many existing papers do exist in the area that should be compared against (e.g ESRNN, Deep factors for forecasting etc) alongside simple econometric ETS models.

     ii. Assuming the paper is tackling models which used frequency-domain-based representations to improve time series forecasting, it should be compared against models such at the FEDformer for consistency.


**Summary Of The Paper:**

The paper proposes a new deep learning architecture for multi-horizon forecasting, which aims to improve temporal representation learning using a bespoke Irregularity Representation Block.

**Summary Of The Review:**

While I think the results show initial promise -- clarity with regards to the model's terminology, motivations and the problem it is tackling would need to be improved before it can be accepted.

---

### Decision · Program_Chairs · 2023-01-20

**Decision:**

Reject

**Justification For Why Not Higher Score:**

While the subject of the paper is interesting, it could benefit from clear motivations, problem formulation and architectural justifications.

**Justification For Why Not Lower Score:**

The subject of modeling residuals in time-series data is interesting

**Metareview: Summary, Strengths And Weaknesses:**

The paper presents a method to model the residuals in time-series forecasting. Although this is an important problem to study, the paper confuses several well-defined terminologies - For eg., residuals have been defined as irregularity and model has been defined as reflections [Reviewer Lsd6].

Authors have not addressed concerns raised by reviewer Lsd6 that raised concerns about the suitability of the architecture used and problem formulation.